# Determinants of anxiety and depression among type 2 diabetes mellitus patients: A hospital-based study in Bangladesh amid the COVID-19 pandemic

## Research Article

anxiety and depressive symptoms; Bangladesh; COVID-19; psychological disorders; type 2 diabetes mellitus

**Corresponding author:**
Suvasish Das Shuvo;
Email: shuvo_nft@just.edu.bd

M.T.H. and S.D.S. authors have made equal contributions to this work.

Md. Toufik Hossen[1], Suvasish Das Shuvo[1] , Sanaullah Mazumdar[1] ,
Md. Sakhawot Hossain[1] , Md. Riazuddin[1] , Deepa Roy[2] ,
Bappa Kumar Mondal[3], Rashida Parvin[1], Dipak Kumar Paul[4] and
Md. Moshiuzzaman Adnan[1]

[1]Department of Nutrition and Food Technology, Jashore University of Science and Technology, Jashore, Bangladesh;
[2]Department of Mathematics, Jashore University of Science and Technology, Jashore, Bangladesh; [3]Department of Food Fortification, Nutrition International, Dhaka, Bangladesh and [4]Department of Applied Nutrition and Food Technology, Islamic University, Kushtia, Bangladesh

## Abstract

Anxiety and depression are common psychological disorders in patients with type 2 diabetes mellitus (T2DM), which was upsurging worldwide amid the COVID-19 pandemic. This study aimed to explore factors associated with anxiety and depression among T2DM patients in Bangladesh during the COVID-19 pandemic. A cross-sectional study was conducted among T2DM patients using face-to-face interviews. Anxiety and depressive symptoms were measured using the CAS and PHQ-9 scales. Outcomes were assessed including sociodemographic, lifestyle, anthropometric, and challenges of getting routine medical and healthcare access-related questions. The prevalence of anxiety and depressive symptoms were 29.8% and 22.7%, respectively. Regression analysis reported that males older than 50 years, illiterate, unemployed or retired, urban residents, below the recommended level of moderate to vigorous physical activity (MVPA), low dietary diversity score (DDS) and obese respondents were associated with higher odds of anxiety and depressive symptoms. Moreover, respondents with transport difficulties, unaffordable medicine, medicine shortages, close friends or family members diagnosed with COVID-19 and financial problems during COVID-19 had higher odds of anxiety and depressive symptoms than their counterparts, respectively. Our study suggests providing psychological support, such as home-based psychological interventions, and highlighting policy implications to ensure the well-being of T2DM patients in Bangladesh during the pandemic.

## Impact statement

The study sheds light on the critical intersection of mental health and chronic illness during a global health crisis. This research identifies significant determinants of anxiety and depression among type 2 diabetes mellitus patients in Bangladesh amid the COVID-19 pandemic. The findings offer valuable insights for healthcare providers, policymakers and researchers, emphasizing the need for integrated care approaches to address the holistic well-being of patients in the context of diabetes management. This approach ultimately leads to improved health outcomes and enhanced quality of life during such challenging times. This study serves as a crucial reference point for future interventions and public health strategies aimed at mitigating the psychological impact of chronic diseases in the backdrop of a pandemic.



## Introduction

COVID-19 was declared a pandemic by the World Health Organization (WHO) on March 11, 2020, based on its global prevalence. It has had an impact on people's lives all around the world due to its rapid spread and high mortality rate (Kim et al., 2022). To reduce the spread of COVID-19, some attempts have been made worldwide, including in Bangladesh, such as nationwide lockdowns, residential or institution-based isolation or quarantine, closing all public activities and educational institutions, and restrictions on social and community mobility (Hosen et al., 2021; Sakib, 2021). As a result, such restrictions and limitations had an impact on daily life, shocking fatality rates in some nations, and media coverage affected public life, and most crucially mental health (Das, 2020; Lima et al., 2020; Sakib, 2021). During the COVID-19 pandemic, mental health

has been a major crisis around the world, particularly for people who were suffering from noncommunicable chronic diseases such as diabetes mellitus, heart disease and hypertension (Gordon Patti and Kohli, 2022). Numerous studies have explored the correlation between diabetes and mental health, particularly depression in the early stages of the pandemic, which adversely impacted the psychological well-being of diabetes patients, including increased suicide ideation one year post-pandemic (Alessi et al., 2020; Joensen et al., 2020; Moradian et al., 2021). Furthermore, a meta-analysis study found the prevalence of anxiety and diabetes distress were 20% and 36%, respectively, among type 2 diabetes mellitus (T2DM) patients (García-Lara et al., 2022).

However, people with chronic depression and anxiety in addition to diabetes are more likely to have diabetes symptoms, experience complications, have an increased disease burden, develop additional comorbidities, engage in inadequate self-care and have an impaired quality of life (Semenkovich et al., 2015). The COVID-19 pandemic also created additional challenges for diabetic patients, including disruptions in routine diabetes care, medication and access to healthcare facilities, which were linked to higher levels of stress, fear of infection and concerns about the accessibility of adequate care in people with T2DM (Grabia et al., 2020; Abdoli et al., 2021; Fekadu et al., 2021; Mohseni et al., 2021). Besides, lack of access to healthy foods, safe locations to exercise and financial stress have exacerbated the harmful effects of diabetes during the pandemic (Abdoli et al., 2021; Alessi et al., 2021; Sujan et al., 2022). Similarly, diabetic patients experiencing anxiety and depression during the COVID-19 pandemic were more likely to engage in unhealthful behaviors, including reduced physical activity, less sleep and increased smoking (Sisman et al., 2021; Kim and Kim, 2022). Moreover, decreasing physical activity levels during the pandemic had a detrimental effect on the glycemic management of diabetic patients (Kim et al., 2022).

Meanwhile, diabetic patients with anxiety and depression are exacerbating the diabetic complications with a double burden and impact on glycemic control (Joensen et al., 2020). Bangladesh is one of the leading country with a higher prevalence of mortality and morbidity of diabetic mellitus in the world (IDF, 2022), so amid such a challenging context, it is imperative to determine and explore the mental health difficulties faced by T2DM patients in Bangladesh during the pandemic. However, several studies already conducted in Bangladesh during the COVID-19 pandemic have provided insight into depression and mental health among college students (Islam et al., 2020), physicians (Hasan et al., 2022) and the general population in Bangladesh, apart from T2DM patients (Al Zubayer et al., 2020; Hosen et al., 2021; Mistry et al., 2021). Evidence proposes that the psychological challenges that might emerge in the aftermath of significant disasters, such as the COVID-19 pandemic, can exhibit varying patterns over time. Conversely, being a low-income nation with a substantial number of individuals affected by type 2 diabetes mellitus, the pandemic's effect on the mental health of this patient remains insufficiently comprehended. Although one study conducted during the COVID-19 pandemic intended to examine the relationship between fasting and diabetes distress and depressive symptoms in individuals with T2DM during Ramadan (a holy month for Muslims) (Sultana et al., 2022), the aim of this study was quite distinguished from our study. As a consequence, it has remained unexplored to investigate the status of psychological health and its predictors among T2DM patients during the COVID-19 in Bangladesh. Therefore, there is a crying need to find out the prevalence of anxiety and depression with its associated factors among diabetic patients in Bangladesh during the COVID-

19 pandemic. Subsequently, our study findings can support the relevant public health policymakers to implement effective policies, programs and insightful actions for this vulnerable T2DM patient in Bangladesh as a mental health issue in such a terrific situation. In this context, this study aims to evaluate the prevalence of anxiety and depression and its associated factors among type 2 diabetes patients in Bangladesh during the COVID-19 pandemic.

## Methods

### Study design and participants

This cross-sectional study was conducted among Bangladeshi T2DM patients from February to March 2022 at the two diabetic hospitals named *Ahad* Diabetic and Health Complex, and *Kapotakkho Lions* Eye and Diabetic Hospital in the Jashore District of Bangladesh. In each of these two hospitals, approximately 50 to 120 patients per day visit for treatment and routine care. Participants were enrolled if the diabetic hospital-attending physician diagnosed type 2 diabetes according to WHO criteria. The diabetic hospital's physician defined diabetes in accordance with WHO criteria, which involved either a fasting plasma glucose (FPG) level equal to or greater than 7.0 mmol/L (126 mg/dL) or a 75 g oral glucose tolerance test (OGTT) (WHO, 2023). We excluded patients with type 1 diabetes, gestational diabetes, those on insulin therapy, and seriously ill patients who were unable to attend the interviews. In order to be eligible, participants needed to meet the following criteria: they had to be adults aged 21 years or older, diagnosed with T2DM, residing in Bangladesh during the COVID-19 outbreak, and capable of comprehending the Bengali language. We employed a simple random sampling method to choose our desired number of respondents. Our sample size was determined using specific parameters, including a 95% confidence level, a 5% margin of error, a 90% test power, and an assumed 95% response rate, all under the assumption of a 50% prevalence rate. However, recent research indicated that the prevalence of anxiety and depressive symptoms among the general population was 33.7% and 57.9%, respectively during the COVID-19 pandemic in Bangladesh (Banna et al., 2022). Given this information, we hypothesized that physicians in Bangladesh might also experience psychological difficulties at a rate of approximately 50%. Accordingly, our initial sample size calculation yielded 384 participants. We factored in a 5% nonresponse rate, leading to a final sample size of 1,036.

### Data collection

The study team conducted data collection through face-to-face interviews using a structured close-ended questionnaire. To ensure consistency, the questionnaire was first translated from English to Bengali and then back-translated to English by two-language experts. Following necessary modifications based on a pretest with 50 participants, the questionnaire was used in the study. For data collection, eight research assistants with experience in conducting health surveys were recruited, and they underwent in-depth training through Zoom meetings before commencing the data collection process.

### Measures

#### Outcome measure

The primary outcomes of the study were anxiety and depressive symptoms. Participants' depressive symptoms were assessed using

the Patient Health Questionnaire-9 (PHQ-9) (Yu et al., 2012). The PHQ-9 has been one of the most convenient tools used in primary healthcare for the diagnosis of depression and yields a major depression diagnosis according to DSM-IV criteria with a continuous severity score. We used the standard cutoff scores with the PHQ-9 to classify as no or minimal (0–4), mild (5–9), and moderate to severe (≥10) symptoms of depression. A total score of ≥10 indicated possible major depression, with a sensitivity of 80% and specificity of 92% (Manea et al., 2012).

The coronavirus anxiety scale (CAS) measured self-reported coronavirus crisis anxiety. Based on two weeks of experiences, each CAS item was assessed on a 5-point scale: 0 = "not at all", 1 = "several days", 2 = "more than half the days", and 3 = "nearly every day". Anxiety was scored from 0–21, with higher scores indicating more anxiety. This scaling format matches the DSM-5 cross-cutting symptom measure. A CAS total score ≥ 9 suggests coronavirus-related anxiety disorder (Lee, 2020). Both scales were also confirmed for validity in the Bengali language (Ahmed et al., 2022; Rahman et al., 2022).

### Explanatory variables

The explanatory variables examined in this study encompassed sociodemographic information, specifically gender, age, education, occupation, monthly income in Bangladeshi Taka (BDT), where 1 USD ~ 90 BDT, residence, lifestyle-related characteristics (including physical activity, physical exercise, smoking habit, duration of diabetes and dietary diversity score (DDS)), anthropometric measurement of weight, height used to compute BMI using the formula weight (kg)/height$^2$ (meter), challenges of getting routine medical and healthcare access-related characteristics (including transport difficulties, limited self-care practice, delayed care seeking, unaffordable medicine, medicine shortage, staff shortage, decreased inpatient capacity, a close friend or family member diagnosed with COVID-19 and financial problem during COVID-19) were calculated (Mohseni et al., 2021; Kim and Kim, 2022; Sultana et al., 2022).

Preexisting medical conditions such as hypertension, hyperlipidemia, coronary artery disease, cerebrovascular illness, cardiovascular difficulties, renal disease, asthma/COPD (chronic obstructive pulmonary disease), and arthritis were collected from patient self-reports (Akter et al., 2014). The participants confirmed the accuracy of this information, following which a certified physician reviewed the prescriptions to validate the diagnoses. Individuals with a history of hypertension, those currently taking anti-hypertensive medication, or those exhibiting elevated blood pressure during the interview were all considered. A diagnosis of hypertension was established based on a systolic blood pressure (SBP) reading of 140 mmHg and a diastolic blood pressure (DBP) reading of 90 mmHg (ICF, 2019).

The respondents' DDSs, assessed by FAO and the FANTA Project using 24-h dietary recall data, were based on the consumption of 12 different food groups, with one point assigned for each group consumed during the reference period (Swindale and Bilinsky, 2006; Kennedy et al., 2011). Respondents who reported consuming items from all food groups during the reference period received a maximum dietary diversity score of 12 points. The overall DDS is on a scale of 0 to 12, categorized into three groups: low (scores of 0–3), moderate (scores of 4–6), and high (≥7) (Kennedy et al., 2011).

### Data analysis

Descriptive analysis was used to evaluate the variables' distributions. Anxiety and depression symptom prevalence across categories was compared using the chi-square test at a 5% level of significance. All relevant factors identified through multivariate analyses were included in binary logistic regression model to determine predictors of anxiety and depressive symptoms after an assessment for collinearity. The statistical significance level was set at p-value <0.05 and adjusted odds ratios (ORs) were reported with a 95% confidence interval (95% CI).

### Results

#### Sociodemographic and healthcare access characteristics

A total of 1,036 T2DM respondents participated in the survey, with 63.5% being female, 44.5% aged between 50 and 64 years, 22.4% illiterate, 65.1% manual workers (work involving the hands, as opposed to an office job), and 59.9% residing in urban areas, respectively. It should be mentioned that 71.7% had moderate to vigorous physical activity (MVPA), 82.2% were nonsmokers, 42.4% were overweight, and 70.0% had uncontrolled diabetes (≥7.0 mmol/L). Notably, 41.4% had comorbidity, and 23.3% had low DDS scores. Regarding the prevalence of anxiety and depression, 29.8% and 22.7% had depressive (PHQ-9 score ≥ 10) and anxiety (CAS-scale ≥9) symptoms, respectively, but most importantly, more than 10.5% had severe depression. Moreover, challenges of getting routine medical and health care, 14.5% had transport difficulties, 36.5% had unaffordable medicine, 22.4% experienced medicine shortage, 14.4% had a close friend or family member diagnosed with COVID-19, and 46.6% faced financial problems during COVID-19 (Table 1).

#### Prevalence of anxiety and depressive symptoms

The study showed that females (24.4%) suffered more from anxiety than males (18.8%). Similarly, depression was more prevalent among females (32.1% vs. 21.7% males). In addition to that, anxiety and depression were more prevalent among unemployed/retired respondents (19.7% and 38.1%) than other respondents and it was found statistically significant. Anxiety and depression were also more prevalent among urban residence patients (23.2% and 31.6%) than among rural residence respondents (Table 2). Furthermore, anxiety was more common among respondents who had a family history of diabetes (27.4%). However, variables including transport difficulties (37.3% and 32.7%), limited self-care practice (32% and 41.4%), unaffordable medicine (25.6% and 34.4%), medicine shortage (40.5% and 34.9%), and financial problem (25.5% and 31.1%) during COVID-19 had a significant relationship with anxiety and depression. The study also showed that respondents' close friends or family members diagnosed with COVID-19 had a higher occurrence of anxiety (Table 2).

#### Factors associated with anxiety and depressive symptoms

Table 3 represents the factors associated with anxiety and depressive symptoms in the binary logistic regression model. Regression analysis found that females were 1.85 times and 2.43 times more likely to be anxious and depressed than male patients. Age between 50 to 64 years had 3.51 times higher odds of anxiety than those below 35 years. However, the presence of anxiety and depression was 2.46 times and 1.79 times higher above age 65 years than the others. In addition to that, respondents who had no education and primary education had higher odds of suffering from anxiety (2.20 times and 2.41 times) and depression (2.36 times and 2.45 times)

**Table 1.** Sociodemographic and healthcare access characteristics of the respondents (*n* = 1,036)

| Variables | Category | Total, *n* (%) |
|---|---|---|
| Sex | Female | 658 (63.5) |
| | Male | 378 (36.5) |
| Age | Below 35 years | 90 (8.7) |
| | 35–49 years | 318 (30.7) |
| | 50–64 years | 461 (44.5) |
| | Above 65 years | 167 (16.1) |
| Education | Graduates | 101 (9.7) |
| | HSC | 87 (8.4) |
| | Secondary | 293 (28.3) |
| | Primary | 323 (31.2) |
| | Illiterate | 232 (22.4) |
| Occupation | Manual worker | 676 (65.1) |
| | Non-manual worker | 168 (9.2) |
| | Unemployment/retired | 192 (18.5) |
| Monthly income | >20,000 BDT | 213 (20.6) |
| | 15,000–20,000 BDT | 330 (31.8) |
| | 10,001–15,000 BDT | 318 (30.7) |
| | <10,000 BDT | 140 (13.5) |
| | Depend on other | 35 (3.4) |
| Residence | Rural | 416 (40.1) |
| | Urban | 620 (59.9) |
| Family history of diabetes mellitus | No | 627 (60.5) |
| | Yes | 409 (39.5) |
| Moderate to vigorous physical activity (MVPA) | Less than the recommended level | 295 (28.3) |
| | Recommended level | 741 (71.7) |
| Physical exercise | Low (<30 min) | 349 (33.7) |
| | High (≥30 min) | 687 (66.3) |
| Smoking habit | Non-smoker | 852 (82.2) |
| | Ex-smoker | 114 (11.0) |
| | Current smoker | 70 (6.8) |
| Duration of diabetes mellitus | Below 6 years | 628 (60.6) |
| | 6–15 years | 319 (30.8) |
| | Above 15 years | 89 (8.6) |
| BMI | Healthy weight (18.5–22.9 kg/m$^2$) | 261 (25.2) |
| | Underweight (<18.5 kg/m$^2$) | 38 (3.7) |
| | Overweight (23–27.5 kg/m$^2$) | 439 (42.4) |
| | Obese (> 27.5 kg/m$^2$) | 298 (28.7) |
| Fasting blood glucose level | Control (<7.0 mmol/L) | 311 (30.0) |
| | Uncontrolled (≥7.0 mmol/L) | 725 (70.0) |
| Comorbidity | No | 607 (58.6) |
| | Yes | 429 (41.4) |
| DDS | High | 318 (30.7) |

(*Continued*)

**Table 1.** (*Continued*)

| Variables | Category | Total, *n* (%) |
|---|---|---|
| | Moderate | 477 (46.0) |
| | Low | 241 (23.3) |
| Depression (PHQ-9 scale) | No Depression | 325 (31.4) |
| | Mild Depression | 402 (38.8) |
| | Moderate Depression | 200 (19.3) |
| | Moderately Severe Depression | 93 (9.0) |
| | Severely Severe Depression | 16 (1.5-) |
| | PHQ-9 score ≥ 10 | 309 (29.8) |
| Anxiety (CAS scale) | No < 9 | 801 (77.3) |
| | Yes ≥ 9 | 235 (22.7) |
| Challenges of getting routine medical and health care | | |
| Transport difficulties | No | 886 (85.5) |
| | Yes | 150 (14.5) |
| Limited self-care practice | No | 908 (87.6) |
| | Yes | 128 (12.4) |
| Delayed care seeking | No | 837 (80.8) |
| | Yes | 199 (19.2) |
| Unaffordable medicine | No | 658 (63.5) |
| | Yes | 378 (36.5) |
| Medicine shortage | No | 804 (77.6) |
| | Yes | 232 (22.4) |
| Staff shortage | No | 937 (90.4) |
| | Yes | 99 (9.6) |
| Decreased inpatient capacity | No | 817(78.8) |
| | Yes | 219 (21.2) |
| Close friend or family member diagnosed with COVID-19 | No | 887 (85.6) |
| | Yes | 149 (14.4) |
| Financial problems during COVID-19 | No | 553 (53.4) |
| | Yes | 483 (46.6) |

than higher educated respondents. Respondents who were non-manual workers and unemployed/retired had higher odds of anxiety (1.25 times and 3.53 times) and depression (2.19 times and 2.95 times) than manual workers. Furthermore, respondents' monthly family income <10,000 BDT were 1.46 times and 1.85 times more likely to suffer from anxiety and depression than their counterparts. Respondents who were residing in urban areas were more likely to have anxiety (2.21 times) and depression (2.81 times). Meanwhile, respondents who had higher physical exercise and recommended levels of MVPA were less likely to have anxiety symptoms (0.49 times and 0.73 times) and depression (0.43 times and 0.79 times). Respondents who were current smokers, obese, and with low DDS were associated with higher odds of anxiety (1.52 times, 1.42 times, and 1.38 times) and depressive symptoms (1.11 times, 1.25 times, and 1.62 times), respectively.

**Table 2.** Distribution of anxiety and depression according to sociodemographic and healthcare access characteristics (*n* = 1,036)

| Variables | Category | Anxiety | | Depression | |
|---|---|---|---|---|---|
| | | Yes, *n* (%) | *P*-value | Yes, *n* (%) | *P*-value |
| Total prevalence | | 235 (22.7) | | 309 (29.8) | |
| Sex | Female | 163 (24.4) | 0.026 | 211 (32.1) | 0.032 |
| | Male | 72 (18.8) | | 98 (21.7) | |
| Age | Below 35 years | 16 (17.7) | 0.293 | 22 (24.4) | 0.487 |
| | 35–49 years | 58 (18.2) | | 92 (28.9) | |
| | 50–64 years | 113 (24.5) | | 139 (30.1) | |
| | Above 65 years | 48 (28.7) | | 56 (33.5) | |
| Education | Graduates | 19 (18.8) | 0.414 | 22 (21.8) | 0.201 |
| | HSC | 21 (24.1) | | 33 (37.9) | |
| | Secondary | 52 (17.7) | | 90 (30.7) | |
| | Primary | 89 (27.5) | | 96 (29.7) | |
| | Illiterate | 54 (23.3) | | 68 (29.3) | |
| Occupation | Manual worker | 120 (17.7) | 0.014 | 176 (26.1) | 0.032 |
| | Non-manual worker | 26 (15.5) | | 60 (35.7) | |
| | Unemployment/retired | 38 (19.7) | | 73 (38.1) | |
| Monthly income | >20,000 BDT | 39 (18.3) | 0.873 | 62 (29.1) | 0.482 |
| | 15,000–20,000 BDT | 77 (23.3) | | 91 (27.5) | |
| | 10,001–15,000 BDT | 70 (22.1) | | 95 (29.8) | |
| | <10,000 BDT | 35 (25.1) | | 51 (36.4) | |
| | Depend on other | 14 (40.0) | | 10 (28.6) | |
| Residence | Rural | 91 (21.8) | 0.052 | 113 (27.2) | 0.013 |
| | Urban | 144 (23.2) | | 196 (31.6) | |
| Family history of diabetes mellitus | No | 133 (21.2) | 0.049 | 195 (31.1) | 0.260 |
| | Yes | 112 (27.4) | | 114 (27.8) | |
| Moderate to vigorous physical activity (MVPA) | Less than the recommended level | 90 (30.5) | 0.421 | 94 (32.1) | 0.317 |
| | Recommended level | 145 (19.5) | | 214 (28.8) | |
| Physical exercise | Low (<30 min) | 109 (31.2) | 0.863 | 109 (31.2) | 0.488 |
| | High (≥30 min) | 126 (18.3) | | 200 (29.1) | |
| Smoking habit | Non-smoker | 150 (17.6) | 0.720 | 259 (30.4) | 0.422 |
| | Ex-smoker | 47 (41.2) | | 28 (24.6) | |
| | Current smoker | 38 (54.2) | | 22 (31.4) | |
| Duration of diabetes mellitus | Below 6 years | 113 (17.9) | 0.210 | 175 (27.8) | 0.370 |
| | 6–15 years | 90 (28.2) | | 106 (33.2) | |
| | Above 15 years | 32 (35.9) | | 27 (31.9) | |
| BMI | Healthy weight (18.5–22.9 kg/m$^2$) | 57 (21.7) | 0.496 | 72 (27.6) | 0.545 |
| | Underweight (<18.5 kg/m$^2$) | 10 (26.3) | | 9 (23.7) | |
| | Overweight (23–27.5 kg/m$^2$) | 96 (21.8) | | 132 (30.1) | |
| | Obese (> 27.5 kg/m$^2$) | 72 (24.1) | | 96 (32.2) | |
| Fasting blood glucose level | Control (<7.0 mmol/L) | 82 (43.4) | 0.585 | 94 (30.2) | 0.854 |
| | Uncontrolled (≥7.0 mmol/L) | 153 (21.1) | | 215 (29.6) | |
| Comorbidity | No | 139 (22.9) | 0.352 | 185 (30.5) | 0.584 |
| | Yes | 96 (22.4) | | 124 (28.9) | |

(*Continued*)

**Table 2.** (*Continued*)

| Variables | Category | Anxiety | | Depression | |
| --- | --- | --- | --- | --- | --- |
| | | Yes, *n* (%) | *P*-value | Yes, *n* (%) | *P*-value |
| DDS | High | 79 (24.8) | 0.115 | 92 (28.9) | 0.891 |
| | Moderate | 99 (20.7) | | 143 (29.9) | |
| | Low | 57 (23.6) | | 74 (30.7) | |
| Challenges of getting routine medical and health care | | | | | |
| Transport difficulties | No | 179 (20.2) | 0.021 | 260 (29.3) | 0.011 |
| | Yes | 56 (37.3) | | 49 (32.7) | |
| Limited self-care practice | No | 182 (20.4) | 0.252 | 268 (29.5) | 0. 373 |
| | Yes | 53 (41.4) | | 41 (32.0) | |
| Delayed care seeking | No | 167 (19.9) | 0.024 | 241 (28.8) | 0.034 |
| | Yes | 68 (34.1) | | 68 (34.2) | |
| Unaffordable medicine | No | 128 (19.5) | 0.046 | 179 (27.2) | 0.016 |
| | Yes | 97 (25.6) | | 130 (34.4) | |
| Medicine shortage | No | 141 (17.5) | 0.012 | 228 (28.4) | 0.035 |
| | Yes | 94 (40.5) | | 81 (34.9) | |
| Staff shortage | No | 196 (20.9) | 0.430 | 282 (30.1) | 0.554 |
| | Yes | 39 (39.4) | | 27 (27.3) | |
| Decreased inpatient capacity | No | 146 (17.8) | 0.094 | 246 (30.1) | 0.701 |
| | Yes | 79 (36.1) | | 63 (28.7) | |
| Close friend or family member diagnosed with COVID-19 | No | 158 (17.8) | <0.001 | 263 (29.6) | 0.766 |
| | Yes | 77 (51.7) | | 46 (30.8) | |
| Financial problems during COVID-19 | No | 102 (18.4) | 0.001 | 159 (28.7) | 0.012 |
| | Yes | 123 (25.5) | | 150 (31.1) | |

BMI, body mass index; DDS, dietary diversity score; HSC, higher secondary school certificate.

In addition, respondents with transport difficulties (1.40 times and 1.87 times), unaffordable medicine (1.81 times and 2.37 times), medicine shortage (1.55 times and 1.35 times), a close friend or family member diagnosed with COVID-19 (3.61 times and 1.61 times), and financial problem during COVID-19 (1.70 times and 2.89 times) had higher odds of anxiety and depressive symptoms than their counterparts, respectively.

## Discussion

Our findings revealed that the prevalence of anxiety and depressive symptoms were 22.7% and 29.8%, respectively. The anxiety prevalence in our study was similar to studies in Saudi Arabia (Magliah et al., 2021). A study in Turkey found the prevalence of anxiety and depression were 46.6% and 39.3%, respectively, which were higher than in the current study (Sisman et al., 2021). The prevalence of depression in the current research was lower than that reported in Brazil and Saudi Arabia (Abdoli et al., 2021). However, methodological differences, varying levels of available resources in each country and the utilization of different scales and cutoff scores in diverse surveys, as well as authentic distinctions, may account for the variances observed between study results. In addition, social and cultural variables influence the occurrence of depression, resulting in varying rates of depression-related illnesses among

nations and across communities and ethnic groups within the same nation (Lloyd et al., 2012).

As in previous studies, we found several parameters related to some subgroups of T2DM patients using regression analysis (Hossain et al., 2020; Sayeed et al., 2020;Al-Sofiani et al., 2021; Hosen et al., 2021). Our study found higher odds of anxiety and depressive symptoms in female respondents compared to their male counterparts. Similar research also indicated that depression and anxiety levels were higher in females than in males (Sayeed et al., 2020; Hosen et al., 2021; Sisman et al., 2021; Kim et al., 2022). Females are more susceptible to anxiety and depression than males as a result of some combination of factors including genetic, hormonal, environmental, societal, and cultural pressures, stress, etc. (Kuehner, 2017). Regarding age, patients aged 50 years or above had higher anxiety and depression than their counterparts. Different researchers exhibited that depression and anxiety symptoms occurred more often with increasing age with diabetic symptoms (Hossain et al., 2020; Al-Sofiani et al., 2021; Kamrul-Hasan et al., 2022; Kim et al., 2022). Diabetes distress in older people could be caused by problems with self-care, other health problems, being disabled, or not having enough social support (Hasan et al., 2022). Additionally, we found respondents with lower educational levels had higher odds of anxiety and depression which is similar to the Bangladeshi study that was conducted among chronic disease patients during COVID-19 (Sayeed et al., 2020). Another study

**Table 3.** Odds of binary logistic regression of predictive study variables with anxiety and depression among type 2 diabetes patients

| Variables | | OR (95% Conf. Interval) | | | |
|---|---|---|---|---|---|
| | | Anxiety | *P*-value | Depression | *P*-value |
| Sex | Male | 1 | | 1 | |
| | Female | **1.85 (1.35–2.45)** | **0.001** | **2.43 (1.77–3.23)** | **0.003** |
| Age | Below 35 years | 1 | | 1 | |
| | 35–49 years | 2.47 (0.89–5.59) | 0.402 | 1.33 (0.73–2.41) | 0.347 |
| | 50–64 years | **3.51 (1.58–7.78)** | **0.002** | **1.40 (1.18–2.83)** | **0.025** |
| | Above 65 years | **2.46 (1.97–6.22)** | **0.056** | **1.79 (1.48–3.62)** | **0.011** |
| Education | Graduates | 1 | | 1 | |
| | HSC | 1.81 (0.78–4.21) | 0.163 | 1.42 (0.75–2.69) | 0.281 |
| | Secondary | 2.07 (0.92–4.64) | 0.076 | 1.63 (0.90–2.97) | 0.104 |
| | Primary | **2.20 (1.38–5.90)** | **0.016** | **2.36 (1.17–4.77)** | **0.001** |
| | Illiterate | **2.41 (1.47–5.45)** | **0.033** | **2.45 (1.78–3.67)** | **0.014** |
| Occupation | Manual worker | 1 | | 1 | |
| | Non-manual worker | **1.25 (1.16–2.61)** | **0.039** | **2.19 (1.83–4.31)** | **0.041** |
| | Unemployment/retired | **3.53 (1.55–6.20)** | **0.010** | **2.95 (1.43–4.12)** | **0.013** |
| Monthly family income | >20,000 BDT | 1 | | 1 | |
| | 15,000–20,000 BDT | 1.20 (0.72–1.98) | 0.472 | 0.94 (0.63–1.41) | 0.789 |
| | 10,001–15,000 BDT | 1.21 (0.73–2.02) | 0.451 | 1.03 (0.68–1.56) | 0.856 |
| | <10,000 BDT | **1.46 (1.25–3.28)** | **0.026** | **1.85 (1.68–3.22)** | **0.001** |
| | Depend on other | 0.68 (0.23–2.02) | 0.494 | 0.86 (0.36–2.04) | 0.736 |
| Residence | Rural | 1 | | 1 | |
| | Urban | **2.21 (1.13–3.61)** | **<0.001** | **2.81 (1.43–3.89)** | **<0.001** |
| Family history of diabetes mellitus | No | 1 | | 1 | |
| | Yes | 1.32 (0.93–1.88) | 0.116 | 0.85 (0.63–1.16) | 0.325 |
| Physical exercise | Low (<30 min) | 1 | | 1 | |
| | High (≥30 min) | **0.49 (0.26–0.94)** | **0.035** | **0.43 (0.24–0.87)** | **0.038** |
| Moderate to vigorous physical activity (MVPA) | Less than the recommended level | 1 | | 1 | |
| | Recommended level | **0.73 (0.54–0.91)** | **0.015** | **0.79 (0.56–0.97)** | **0.028** |
| Smoking habit | No-smoker | 1 | | 1 | |
| | Ex-smoker | 1.20 (0.60–2.42) | 0.146 | 0.48 (0.27–0.85) | 0.013 |
| | Current smoker | **1.52 (1.25–3.54)** | **0.032** | **1.11 (1.05–2.14)** | **0.047** |
| Duration of diabetes mellitus | Below 6 years | 1 | | 1 | |
| | 6–15 years | 1.05 (0.72–1.53) | 0.792 | 1.34 (0.98–1.84) | 0.066 |
| | Above 15 years | 0.60 (0.28–1.29) | 0.196 | 1.18 (0.68–2.04) | 0.543 |
| BMI | Healthy weight (18.5–22.9 kg/m$^2$) | 1 | | | |
| | Underweight (<18.5 kg/m$^2$) | 0.61 (0.24–1.55) | 0.300 | 0.71(0.31–1.65) | 0.439 |
| | Overweight (23–27.5 kg/m$^2$) | 0.50 (0.33–0.77) | 0.265 | 1.09 (0.76–1.56) | 0.629 |
| | Obese (>27.5 kg/m$^2$) | **1.42 (1.26–2.68)** | **<0.001** | **1.25 (1.14–1.85)** | **<0.001** |
| Fasting blood glucose level | Controlled (<7.0 mmol/L) | 1 | | | |
| | Uncontrolled (≥7.0 mmol/L) | 0.76 (0.52–1.12) | 0.178 | 0.85 (0.61–1.17) | 0.330 |
| DDS | High | 1 | | 1 | |
| | Moderate | 0.93 (0.63–1.39) | 0.751 | 1.14 (0.75–1.72) | 0.647 |
| | Low | **1.38 (1.21–1.74)** | **0.002** | **1.62 (1.32–3.22)** | **0.002** |

(*Continued*)

**Table 3.** (*Continued*)

| Variables | | OR (95% Conf. Interval) | | | |
|---|---|---|---|---|---|
| | | Anxiety | *P*-value | Depression | *P*-value |
| Comorbidity | No | 1 | | 1 | |
| | Yes | 0.73 (0.51–1.05) | 0.097 | 0.83 (0.62–1.13) | 0.249 |
| Challenges of getting routine medical and health care | | | | | |
| Transport difficulties | No | 1 | | 1 | |
| | Yes | **1.40 (1.24–2.34)** | **0.041** | **1.87 (1.56–3.35)** | **0.026** |
| Limited self-care practice | No | 1 | | 1 | |
| | Yes | 1.11 (0.61–2.04) | 0.717 | 0.97 (0.59–1.61) | 0.932 |
| Delayed care seeking | No | 1 | | 1 | |
| | Yes | 0.88 (0.52–1.48) | 0.638 | 1.32 (0.88–1.98) | 0.173 |
| Unaffordable medicine | No | 1 | | 1 | |
| | Yes | **1.81 (1.51–3.29)** | **0.001** | **2.37 (1.95–3.47)** | **>0.001** |
| Medicine shortage | No | 1 | | 1 | |
| | Yes | **1.55 (1.43–2.41)** | **0.004** | **1.35 (1.33–2.26)** | **0.001** |
| Staff shortage | No | 1 | | 1 | |
| | Yes | 0.54 (0.28–1.04) | 0.066 | 0.83 (0.49–1.43) | 0.523 |
| Decreased inpatient capacity | No | 1 | | 1 | |
| | Yes | 1.34 (0.77–2.33) | 0.287 | 0.97 (0.60–1.56) | 0.906 |
| Close friend or family member diagnosed with COVID-19 | No | 1 | | 1 | |
| | Yes | **3.61 (2.01–6.48)** | **<0.001** | **1.61 (1.44–2.75)** | **0.027** |
| Financial problems during COVID-19 | No | 1 | | 1 | |
| | Yes | **1.70 (1.47–2.70)** | **0.022** | **2.89 (1.62–3.27)** | **0.028** |

*Note*: Values in bold are statistically significant at *P* < 0.05.

in India showed that low levels of education increased the risk of major depressive disorder (Avasthi et al., 2015). Besides, the perspective of monthly income, patients with lower income had higher odds of anxiety and depression. Our findings are in line with recent studies in other countries that reported a significant association between depression and monthly income (Al-Sofiani et al., 2021; Hosen et al., 2021). A scarcity of financial resources may result in reduced access to healthcare, exacerbating emotional challenges such as anxiety and depression among individuals with low income. The financial strains and mental health concerns can further intensify if they face job loss or reduced work hours (Chatterji et al., 2021; De Miquel et al., 2022). We also found that patients' occupation was significantly associated with anxiety and depression. Patients who had lost their jobs, retired, and nonmanual workers were highly associated with anxiety and depression. Our finding is consistent with different previous research (Palizgir et al., 2013; Hossain et al., 2020). Job loss is a significant contributor to anxiety and depression, particularly among the unemployed, who may also experience financial burdens, exacerbating these mental health issues (Chatterji et al., 2021; De Miquel et al., 2022).

We also found that patients who resided in an urban region had a significantly higher risk of anxiety and depressive symptoms than rural patients. A recent study also showed that urban diabetic patients had more significant anxiety problems (Leong Bin Abdullah et al., 2021). Strict lockdown measures in urban areas, including movement restrictions, transportation difficulties, safety precautions in government and nongovernment offices

and organizations, and others, may increase the risk of anxiety and depression (Das, 2020; Sakib, 2021). Another noteworthy finding was that respondents who had MVPA and higher levels of physical exercise had lower odds of anxiety and depressive symptoms. Our findings are also consistent with previous studies (Nanayakkara et al., 2018; Kandola et al., 2020). Possible mechanisms linking regular physical exercise and MVPA to reduce the anxiety and depression include the elicitation of endorphins, the initiation of thermogenesis, the activation of the mTOR axis in certain brain regions, and the discharge of neurotransmitters like dopamine and serotonin (Silva et al., 2020).

Moreover, respondents who were current smokers and obese had higher odds of anxiety and depressive symptoms than their counterparts. Similar studies have found that smoking and obesity are highly associated with increased risks of diabetes-related mental health issues (Al-Sofiani et al., 2021; Sisman et al., 2021). This could be due to the neuro-biochemical response mechanism of nicotine and monoamine oxidase, which explains the effect of smoking on sadness and anxiety. Persistent smoking impacts depression and anxiety in adolescence because it increases the quantity of nicotine in the blood and reduces the amount of monoamine oxidase, which is strongly associated with depression (Byeon, 2015). Besides, obesity and mental disorders may be characterized by low self-esteem, unhealthy eating patterns, binge eating, and reduced physical activity. Also, high levels of inflammatory biomarkers may lead to depression among obese people (Heidari-Beni et al., 2021). According to a previous study we observed that a less diverse diet was

related to an increased risk of anxiety and depression (Grabia et al., 2020). This is because symptoms of anxiety, such as loss of appetite, can make it difficult to maintain a healthy diet. Moreover, individuals with a less varied diet are more likely to experience higher levels of anxiety and depression according to research (Su et al., 2015).

In our study, the challenges of getting routine medical and health care during COVID-19 were identified as significant risk factors for developing anxiety and depression in T2DM patients. We found that patients with transportation difficulties and medicine shortages were significantly more likely to suffer from anxiety and depressive symptoms. These findings were in line with reports from previous studies during the pandemic (Sayeed et al., 2020; Abdoli et al., 2021; Fekadu et al., 2021). Transportation issues caused by the COVID-19 pandemic have resulted in severe declines in mental health for patients with diabetes, as some were unable to access the medical care they need regularly, causing worry and stress levels to increase (Abdoli et al., 2021; Fekadu et al., 2021). For pharmaceutical shortages, the possible explanation is that diabetic patients who depend on these drugs to manage their condition may experience worry and anxiety, as uncontrolled blood sugar levels and the possibility of developing complications are concerning for individuals with diabetes. Interruptions to the continuity of care due to medication shortages can exacerbate these worries (Palizgir et al., 2013; Mohseni et al., 2021). Furthermore, we found that respondents' close friends or family members diagnosed with COVID-19 were associated with higher odds of anxiety and depression. A similar study showed that fear of getting an infection of COVID-19 is also associated with higher diabetes distress (Sayeed et al., 2020; Kim et al., 2022). Another research in Bangladesh indicated that those who reported being emotionally affected by the epidemic, such as through contact with infected relatives or friends, had greater levels of anxiety and depressive symptoms (Siddique et al., 2021). The link between COVID-19 and a heightened sense of concern for the health and safety of loved ones, particularly those who were vulnerable to serious illness, might be attributed to this factor (Hossain et al., 2020). Financial difficulties during COVID-19 were more prone to the development of anxiety and depression, which was also found in our present study. A similar study also reported that financial crisis patients had higher diabetes distress (Sujan et al., 2022). Various reasons can lead to financial stress in an individual's life, such as unemployment, reduced income, or mounting debt. However, individuals with chronic conditions who are already grappling with the economic consequences of their illness may be more susceptible to experiencing detrimental mental health outcomes, associated with financial strain (Siddique et al., 2021; Sujan et al., 2022).

### Policy implications

This study presents preliminary data on the effects of COVID-19 on mental health among T2DM patients. Longitudinal research is required to comprehend the mental health trajectory. The findings of this study support the implementation of community-based screening, diagnosis, and treatment programs for anxiety and depression. Moreover, these findings may also help stakeholders, governments, and international organizations in building psychological first-aid courses to meet this underserved population's mental health needs. Mobility restrictions during the lockdown should compromise the care of T2DM patients while maintaining any pandemic-related health safety issues since they have disrupted follow-up appointments, limited medication access, and regulated the higher cost of diabetes medications and other equipment. A healthy daily routine, paying attention to needs and feelings,

engaging in recreational and healthy activities (such as yoga and meditation), avoiding rumors that may cause worry and tension and maintaining social networks are just a few of the methods that should be considered.

### Strengths and weaknesses

This study has some major strengths. This is the first study providing insight into prevalent of anxiety and depressive symptoms particularly among Bangladeshi T2DM patients amid the COVID-19 pandemic. Secondly, the study used a face-to-face survey to identify the respondents' sociodemographics, lifestyle, and challenges of getting routine medical and healthcare-related characteristics as confounders that are associated with mental health. In contrast, the high sample size attained during the pandemic indicated the strength of this study, as we had a substantial ability to test our hypotheses. However, we cannot overlook the limitations that should be considered when interpreting this study's findings. First, all of the outcomes were self-reported, which might lead to recall bias. Second, the cross-sectional nature of the study design could not establish a causal relationship between the dependent and independent variables. Third, the findings in the present study might not be generalizable to other populations. Fourth, factors such as the prevalence of COVID-19 and different mortality rates might affect the impacts of COVID-19 on mental health. Finally, we did not collect data about preexisting diagnoses of anxiety and depression.

### Conclusion

The COVID-19 pandemic was placing extra strain on individuals' mental well-being. Findings from the current research reveal that, since the onset of the COVID-19 outbreak in Bangladesh, a significant number of T2DM participants were experiencing symptoms of anxiety and depression. This research also suggests that the factors associated with COVID-19 significantly impacted the mental well-being of individuals. To be more precise, the patients who were male, older, lived in urban areas, had lower educational levels, performed manual labor, were unemployed, had a low income, smoked cigarettes, had fewer varieties in their diet or did not engage in moderate to vigorous exercise had an increased risk of developing depression or anxiety. Unaffordable and scarce medicine, financial stress, and close friends or family members with COVID-19 were increased risk factors for anxiety and depression during the third wave of the epidemic. The results show that authorities need to focus on a time-focused strategy for mental health screening, counseling, and incorporating vigilant monitoring to effectively handle future outbreaks and foster mental well-being among T2DM patients.

### Abbreviations

| | |
|---|---|
| BDT | Bangladeshi Taka |
| BMI | Body Mass Index |
| CAS | Coronavirus Anxiety Scale |
| DDS | Dietary Diversity Score |
| MVPA | Moderate to Vigorous Physical Activity |
| OR | Odds Ratios |
| WHO | World Health Organization |

**Open peer review.** To view the open peer review materials for this article, please visit http://doi.org/10.1017/gmh.2024.8.

**Data availability statement.** There are some restrictions on this data, and it is not available to the public. Data will be available upon reasonable request.

**Acknowledgements.** All the authors wish to express their gratitude to the participants who volunteered for this study.

**Author contribution.** Suvasish Das Shuvo and Md. Toufik Hossen conceptualized the study, synthesized the analysis plan, and conducted the statistical analysis. Suvasish Das Shuvo, Md. Toufik Hossen and Sanaullah Mazumdar compiled the data and interpreted the findings. Md. Toufik Hossen, Suvasish Das Shuvo, Sanaullah Mazumdar, Md. Sakhawot Hossain, Md. Riazuddin, Deepa Roy, Bappa Kumar Mondal, Rashida Parvin, Dipak Kumar Paul, Md. Moshiuzzaman Adnan drafted the manuscript, critically reviewed the manuscript and approved the manuscript.

**Financial support.** This research received no specific grant from any funding agency in the public, commercial, or not-for-profit sectors.

**Competing interest.** The authors declare none.

**Ethics statement.** The authors assert that all procedures contributing to this work comply with the ethical standards of the relevant national and institutional committees on human experimentation and with the Helsinki Declaration of 1975, as revised in 2008. All procedures involving research study participants were approved by the Ethical Review Committee of the Faculty of Biological Science and Technology, Jashore University of Science and Technology (Ref: ERC/FBST/JUST/2022–107). All respondents provided both their written and verbal informed assent. The confidentiality of all information provided by respondents was ensured. The respondents were informed that they could stop the interview at any time.

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
