## [Reviewer Report]

14/9/2023

The Editor

Journal of Global Mental Health 

Dear Sir, 

It is with pleasure that we are submitting our manuscript entitled “Determinants of Anxiety and Depression among Type-2 Diabetes Mellitus Patients: A Hospital-based Study in Bangladesh amid the COVID-19 Pandemic” for consideration to Global Mental Health Journal. This manuscript provides original work which is not under consideration for publication elsewhere. All authors approved the manuscript and this submission. 

The COVID-19 pandemic has led to significant psychological distress worldwide, including among individuals with chronic illnesses such as type-2 diabetes. Our study aimed to assess the prevalence of anxiety and depression related to COVID-19 among type-2 diabetes patients in Bangladesh and to identify associated factors.

We conducted a cross-sectional study among 1036 type-2 diabetes patients in Bangladesh using standardized questionnaires to assess COVID-19 related anxiety and depression. Our results showed that the prevalence of anxiety and depression related to COVID-19 was high among type-2 diabetes patients in Bangladesh. We found that socio-demographic, lifestyle, and anthropometric, challenges of getting routine medical and healthcare-related characteristics were significantly associated with increased anxiety and depression related to COVID-19 among these patients.

Our study adds to the growing body of literature on the mental health impact of the COVID-19 pandemic and its associated factors among vulnerable populations such as individuals with type-2 diabetes. Our findings have important implications for healthcare providers and policymakers in Bangladesh and other low- and middle-income countries facing similar challenges. 

We author feel “Global Mental Health Journal” is the perfect scientific platform to disseminate the findings all over the world. 

Thank you very much for considering our work for publication. We look forward to hearing from you. 

Thank you for your consideration! 

Regards, 

Suvasish Das Shuvo

(The corresponding author)

---

## [Reviewer Report]

Review Comments

Manuscript ID: GMH-23-0188

Research Title: Determinants of Anxiety and Depression among Type-2 Diabetes Mellitus Patients: A Hospital-based Study in Bangladesh amid the COVID-19 Pandemic

The above title research is an interesting and important scientific evidence. Overall, it is well-presented manuscript with appropriate statistical analysis. However, this paper needs substantial improvement in writing (comments below).

Abstract

1. L5, Anxiety and depression are common psychological disorders in patients with type-2 diabetes mellitus (T2DM), which is up-surging worldwide amid the ongoing COVID-19 pandemic.

The COVID-19 is no longer an ongoing phenomenon now. Needs to rewrite the sentence.

2. L7, Use short form of type-2 diabetes mellitus.

3. L21, …… highlighting policy implications to ensure the well-being of T2DM patients in Bangladesh during the pandemic. Not clear what does it mean. Do the authors mean policy action?

Introduction

Introduction section needs massive improvement in terms of setting the context. There are adequate recent citation however they are not coherently arranged. The literature review part requires pointing out the main key message from previous literature. No clear research gap is mentioned neither the contribution of this study.

The rationale for the study is well explained, emphasizing the need to understand the mental health challenges faced by T2DM patients in Bangladesh during the pandemic. However, consider clarifying how this study contributes to the existing knowledge and what novel insights it brings.

Different citation pattern is found in the manuscript. The citation and reference style should be standard in format and as per journal guideline. For example,

1. p3, L43, M. J. Kim et al., 2022 the citation <M. J. Kim et al., 2022.> does not follow proper style.

2. P3, L43-47: To reduce the spread of COVID-19 some attempts have been made worldwide such as nationwide lockdowns, residential or institution-based isolation or quarantine, closing all public activities and educational institutions, and restrictions on social and community mobility (Hosen et al., 2021).

The claim of worldwide measures to stop spread pf COVID-19 in this sentence requires appropriate citation. Hosen et al. 2021 only covers the studies conducted in a single country Bangladesh.

3. P4, L75-76, citation needs to be in proper style.

4. P4, L-78, citation needs to be in proper style.

5. P4, L79-84, A very big sentence however it does have grammatical mistake and inadequacy to express main context.

6. P4, L84-88, What is the insight from these studies? Can the authors highlight their own contribution over the previous studies. The sentence requires grammatical correction.

Methods

1. P5, L114, delete the elaboration Type-2 Diabetes Mellitus (T2DM) as it has been already mentioned in Abstract.

2. P5, L106, “….. approximately 50 to 120 patients visit per day receive treatment and routine care.” needs grammatical correction.

3. L110, punctuation problem.

4. L120, “…. Banna et al. in 2020 indicated….. (Banna et al., 2022)” double citation of the same source? Needs correction.

5. L162, citation needs correction.

6. L176-178, A binary logistic regression model was used to determine predictors of the dependent variables, anxiety and depression symptoms after an assessment for collinearity. Author may write the regression model explicitly in this section with defined dependent and independent variable.

7. L174, STATA, version 14.0, was used for the analysis. May delete this sentence.

Results

1. The authors have described almost all estimated numbers in the text. However, reporting of statistic such as P-value, confidence interval and others information reduces the readability of the paragraph. This is simply repetition of same information as they are already reported in the table. May the authors exclude the repetition and more comprehend the insights of the estimated result.

For example,

a) Section 3.2, The study showed that females (24.4%) suffered more from anxiety than males (18.8%) which was statistically significant (p<0.05).

b) Regression analysis found that females were 1.85 times and 2.43 times more likely to be anxious (OR: 1.85; 95% CI: 1.35-2.45) and, depressed (OR: 2.43; 95% CI: 1.77-3.23) than male patients.

This comment applies to all relevant parts.

2. L263, According to Lloyd et al., social and cultural variables impact the occurrence of depression, resulting in varying rates of depression-related illnesses among nations and across communities and ethnic groups within the same nation (Lloyd et al., 2012).

Repetition of same reference

3. L267, L277-279, L288 (Al-Sofiani, M. E., Albunyan, S., Alguwaihes, A. M., Kalyani, R. R., Golden, S. H., & Alfadda, 2021) Incorrect citation style.

4. L302, incorrection citation style

5. L317, incorrect citation style. This single article is cited in multiple places. Al-Sofiani, M. E., Albunyan, S., Alguwaihes, A. M., Kalyani, R. R., Golden, S. H., & Alfadda, 2021

6. L327, incorrect citation style

7. L332, incorrect citation style

8. L338, incorrect citation style

9. L342, incorrect citation style

10. L352, incorrect citation style

11. L355, incorrect citation style

12. L361, incorrect citation style

13. L367-3L69 incorrect citation style

Policy implication

The paper needs improvement in the policy implications section. The policy implications provided in the discussion are valuable. However, it would be beneficial to summarize these implications and provide clear, actionable recommendations for policymakers. This will help make the practical relevance of the study more apparent.

Conclusion

The conclusion section needs improvement. This section should summarize the main findings and their broader implications for research and public health policy. Reiterate the key takeaway points and highlight how the study contributes to addressing the mental health challenges faced by T2DM patients during the COVID-19 pandemic in Bangladesh.

Other comments

1. L419-L421, make a list of these abbreviations.

2. L423-L425, The abbreviation of author’s names has been used however it has not been used appropriately. Suggest using full name as name abbreviation looks bit unfamiliar in academic publishing.

Overall comments:

Overall, the manuscript provides valuable insights into the mental health challenges faced by T2DM patients during the COVID-19 pandemic in Bangladesh. As described, the article is well conducted with clear methodology and objective. However, this article needs substantial improvement in terms of setting the research context, identifying clear research gap and explaining results with feasible policy implications. The paper also needs a thorough grammatical check. All references need to be properly cited in the text and listed as per the journal guideline. Authors need to make sure all the cited references are listed appropriately.

Should the authors address the comments outlined above, I recommend accepting the paper for publication.

---

## [Reviewer Report]

The author did not mention whether it was an open-ended questionnaire (Line number 128).

Is the validity and reliability of the PHQ-9 and CAS consistently verified in any other studies in the Bangladeshi population? If yes, should mention some reference as an example (Line number 136).

Should be written as 0-21 (Line number 148)

Briefly describe DDS as there is no explanation about this in the methods section (Line number 152)

The description under Respondent’s characteristics should be rewritten as it looks clumsy (Line number 183).

Rewrite the results in line number 209.

Clarify the word manual work (line number 230)

Reference should be checked Das, n.d. (line number 306)

Repetition Anxiety and depressive symptoms (line number 314).

---

## [Reviewer Report]

This paper sets out to address anxiety, depression in type 2 diabetes during the Covid-19. The authors did well on literature review in the introduction, on the methodology and the results. However, of concern, there are substantial references in the discussion that need be taken back to where they belong i.e. under literature review in the introduction. The authors could then in their discussion make references to what was found in the literature review.

---

## [Reviewer Report]

14/12/2023

The Editor

Journal of Global Mental Health 

Dear Sir, 

It is with pleasure that we are submitting our manuscript entitled “Determinants of Anxiety and Depression among Type-2 Diabetes Mellitus Patients: A Hospital-based Study in Bangladesh amid the COVID-19 Pandemic” for consideration to Global Mental Health Journal. This manuscript provides original work which is not under consideration for publication elsewhere. All authors approved the manuscript and this submission. 

The COVID-19 pandemic has led to significant psychological distress worldwide, including among individuals with chronic illnesses such as type-2 diabetes. Our study aimed to assess the prevalence of anxiety and depression related to COVID-19 among type-2 diabetes patients in Bangladesh and to identify associated factors.

We conducted a cross-sectional study among 1036 type-2 diabetes patients in Bangladesh using standardized questionnaires to assess COVID-19 related anxiety and depression. Our results showed that the prevalence of anxiety and depression related to COVID-19 was high among type-2 diabetes patients in Bangladesh. We found that socio-demographic, lifestyle, and anthropometric, challenges of getting routine medical and healthcare-related characteristics were significantly associated with increased anxiety and depression related to COVID-19 among these patients.

Our study adds to the growing body of literature on the mental health impact of the COVID-19 pandemic and its associated factors among vulnerable populations such as individuals with type-2 diabetes. Our findings have important implications for healthcare providers and policymakers in Bangladesh and other low- and middle-income countries facing similar challenges. 

We author feel “Global Mental Health Journal” is the perfect scientific platform to disseminate the findings all over the world. 

Thank you very much for considering our work for publication. We look forward to hearing from you. 

Thank you for your consideration! 

Regards, 

Suvasish Das Shuvo

(The corresponding author)

---

## [Reviewer Report]

I have read the revised copy and noted all the revisions that they made. 

Apart from this, the paper addresses mental health (anxiety and depression), a physical co-morbidity (Type 2 Diabetes Mellitus) in the context of a pandemic (COVID-19). They looked at various social-demographic variables, various physical co-morbidities and some health system problems. The methods are well described, the results well-presented and a fairly detailed discussion. 

In my view, they make original contribution, emanating from a LMIC and in the process contributed to global database on the subject.

I recommend publication.